# Rural infrastructure and poverty in China

**Xiaodi Qin, Haitao Wu\*, Tiecheng Shan**

School of Business Administration, Zhongnan University of Economics and Law, Wuhan, China

\* wuhan_haitao@aliyun.com

## Abstract

The study develops a theoretical framework of how irrigation and drainage infrastructure and rural transportation infrastructure influence poverty. Using panel data on 31 provinces in China from 2002 to 2017, this paper estimates basic and continuous difference-in-differences (DID) models to investigate the preliminary impact of irrigation and drainage infrastructure and rural transportation infrastructure on poverty and further explores the influence mechanisms of these rural infrastructures on poverty by using the mediating effect model. The results show that irrigation and drainage facilities infrastructure can directly reduce poverty. On the one hand, rural transportation infrastructure directly leads to rural hollowing out and aggravates rural poverty; on the other hand, it indirectly promotes poverty reduction by stimulating economic growth. Overall, the positive and negative effects of rural transportation infrastructure on poverty offset each other.

**Data Availability Statement:** All relevant data are within the paper and its Supporting information files and are available from National Bureau of Statistics of China (http://www.stats.gov.cn/), China Statistical Yearbook (https://data.cnki.net/yearbook/Single/N2021110004), and China Rural

## Introduction

Poverty reduction has always been a core component of the Sustainable Development Goals of the United Nations, which are deeply important for promoting the sustainable development of the world especially when facing the threat of normalizing extreme weather. On November 23rd 2020, China announced that it had eliminated absolute poverty nationwide by uplifting all its citizens beyond its set 2300 yuan per year, or around 1.52 dollars per day poverty line. Over the past 40 years, China has pursued many stimuluses to achieve this goal and more than 700 million people has been lifted out of poverty. One of the stimuluses is to promote a variety of infrastructure projects [1]. In fact, infrastructure plays a fundamental role in promoting growth and alleviating poverty in China, especially in rural areas greatly affected by extreme weather. Agriculture, rural areas and farmers' issues have always been China's top priorities since 2004 and there have been 17 No. 1 central documents, made by the government of China, prioritizing development of agriculture and rural areas with different key themes each year. In 2010, China's No. 1 central document proposed to strengthen construction of rural infrastructure, especially for rural irrigation and drainage and transportation infrastructure, to reduce poverty. The central and local government funds as well as national debt funds needed to be invested to the construction and management of rural infrastructure. More specifically, the government arranged 818.3 billion yuan for agriculture, peasants, and rural areas in 2010. Among them, 86.2 billion yuan was allocated for small-sized irrigation projects and 132.3 yuan for rural roads and other public transport infrastructures, beyond the previous investment.

Statistical Yearbook (https://data.cnki.net/yearbook/Single/N2021120010).

**Funding:** This research was funded by National Social Science Foundation of China(grant number: 19ZDA151) and National Natural Science Foundation of China (grant number: 71573277; 71273281). They played an important role in study design and data collection and analysis. Besides, we also appreciate the help of Cunhong Tu, who sponsored the preparation of the manuscript with the fund from the Fundamental Research Funds for the Central Universities (grant number: 202211001).

**Competing interests:** NO authors have competing interests.

After 2010, China's No. 1 central document still attaches great importance to rural infrastructure of irrigation and transportation and keep investing to rural infrastructure to help rural poor people to deal with extreme weather.

Given that China has invested heavily to irrigation and transportation infrastructure especially in rural areas and the miracle of poverty reduction in the past 40 years, three important issues arise and needs to be figured out. First, did the investment to irrigation and transportation infrastructure in 2010 reduce poverty in practice? Second, what roles did irrigation and transportation infrastructure play in influencing poverty, respectively? Third, what is the mechanism behind the impact? Some studies maintain that rural transportation infrastructure, may not have social benefits and succeed in reducing poverty if rural laborers choose to transfer to urban areas due to the lack of agglomeration of local economies [2]. On the contrary, rural poverty may be deepened because of the loss of young and middle-aged rural labor force and the decline of abandoned villages, which is called siphon effect [3]. Moreover, irrigation and drainage infrastructure may also fail to count in that case. Dealing with these three problems can help China's government reasonably adjust investment structure in rural infrastructure and consolidate the achievements on poverty reduction, as well as provide experience for those developing economies troubled by poverty. In addition, under the threat of extreme weather, it is of particular importance to clarify the role of irrigation and drainage infrastructure in affecting poverty. Irrigation and drainage infrastructure, due to the characteristics of relatively lower profits and slower return, usually attracts much less investment than rural transportation infrastructure.

Given the above discussions, this research aims to evaluate the impact and mechanisms of rural irrigation and transportation infrastructure on poverty and makes innovations in several ways. First, this paper theoretically clarifies how irrigation and drainage infrastructure and rural transportation can affect poverty. We supplement theoretical literature concerning how irrigation and drainage infrastructure influences poverty and the underlying mechanism, which are rare in previous research. In addition, we add to a growing body of literature on how rural transportation influences poverty, focusing not on the poverty-reducing effect but also on the poverty-aggravating effect. Second, this paper helps enrich empirical evidence on how rural irrigation and transportation infrastructure influences poverty by applying the basic and continuous DID models and the mediating effect model. We use updated data to make the results more convincible and two-stage least square method to handle endogeneity problem.

The rest of this paper is structured as follows. The next section reviews previous literature. The Framework and Hypotheses section analyzes the theoretical influencing mechanisms of the three kinds of rural infrastructure on poverty. The Materials and Methods section discusses materials and methods. The Results and Conclusion section present the empirical test and discuss the results, respectively. The last section concludes with policy implications, limitations and future research.

## Literature review

Many empirical studies have suggested a positive role for rural infrastructure in allowing improvements to the quality of life of those living in poverty and alleviating poverty. Most of these studies highlight the impact of rural infrastructure on economic growth and thus indirectly on poverty, which is called a trickle-down effect [4–9]. The research by these authors illustrates that rural infrastructure such as rural transportation infrastructure can stimulate economic growth through gains in productivity, which in turn leads to increases in income and poverty alleviation. Sasmal, R., and Sasmal, J. (2016) and Chotia et al.,(2017) examine the

connection between economic growth and poverty alleviation as well as how the two are connected to public infrastructure [10, 11]. The results reveal that economic growth may drive poverty reduction, and infrastructure features largely in both growing the economy and reducing poverty.

Some seminal works also pay attention to how rural infrastructure directly affects poverty. Most of the authors believe that rural infrastructure construction can directly reduce farmers' production costs and improve productivity, thereby increasing their income and reducing poverty [12, 13]. Fan et al. (2005) reveal that investment in rural infrastructure can increase household income and pro-mote poverty reduction, with road infrastructure playing the most important role [6]. More recently, some scholars have pointed out that the improvement of rural infrastructure, especially rural transportation infrastructure, may promote urbanization. However, it may also lead to less productive capital and skilled labor in rural areas. In such "hollow villages", the rural poor may fall into greater poverty [14]. This increase in poverty may offset the poverty reduction effect of rural infrastructure, which we can call the masking effect, often used in mediation analysis in psychology [15–17]. Other studies also explore the relationship between irrigation and drainage infrastructure and rural income or rural poverty [18].

Overall, previous studies have emphasized the importance of rural infrastructure for poverty reduction. Most studies focus on how rural transportation infrastructure affects poverty reduction and usually find the negative relationship between rural transportation infrastructure and poverty. However, few literatures manage to figure out the mechanisms of how the irrigation and drainage infrastructure influence poverty. The specific poverty reduction mechanism behind the irrigation and drainage infrastructure needs to be further determined. In addition, it is important to discuss the potential negative impact of rural transportation infrastructure on poverty reduction and make a comparison with the irrigation and drainage infrastructure. Therefore, this paper contributes to a growing but inconclusive body of literature by theoretically clarifying how irrigation and drainage infrastructure and rural transportation can affect poverty. In addition, this paper uses the basic and continuous DID models to study the impact of different rural infrastructures on poverty. This continuous DID model can solve certain endogeneity problems in the model. The two-stage least squares method and a change in the dependent variables are also used as tests of the robustness of the conclusion. Third, this paper uses the mediating effect model to further explore the mechanisms behind the effects of irrigation and drainage infrastructure and transportation infrastructure on poverty. Different from most existing studies, this study finds that rural transportation infrastructure both aggravates rural poverty and reduces poverty by promoting economic growth. Given the current policy situation and this realistic background, it is necessary to pay more attention to the irrigation and drainage infrastructure, which is related to agriculture production and peasants' life. This consideration will further consolidate the gains in poverty alleviation and prevent future increases in poverty.

## Framework and hypotheses

### Irrigation and drainage infrastructure and poverty

On the one hand, irrigation and drainage infrastructure can directly reduce poverty. As one of the most basic public goods in rural areas, irrigation and drainage infrastructure can improve agricultural production conditions and grain yield by improving irrigation capacity. Farmers are able to adjust their crop structure, develop large-scale breeding programs and engage in processing and nonagricultural industries to eliminate poverty. At the same time, the improvement of irrigation and drainage infrastructure can enhance farmers' ability to deal with

disasters and reduce risks. Therefore, production efficiency will rise with lower agricultural production costs, thereby reducing poverty. In addition, irrigation and drainage infrastructure can release part of the rural labor force from farm work and optimize the work-time structure. This saved labor and time can be used for higher income activities. Thus, poverty can be reduced. On the other hand, irrigation and drainage infrastructure can indirectly reduce poverty by promoting the growth of agriculture, forestry, animal husbandry and fisheries [19]. According to the Cobb-Douglas production function, the elasticity of capital and labor may directly determine growth. Irrigation and drainage infrastructure can increase the output from a unit of capital and labor by reducing the impact of floods and other disasters on them. The growth in agriculture, forestry, animal husbandry and fisheries can continue, thus indirectly affecting poverty.

Based on the literature review, we have the following hypotheses:

Hypothesis 1a. Irrigation and drainage infrastructure can directly reduce poverty.

Hypothesis 1b. Irrigation and drainage infrastructure can indirectly reduce poverty through increasing the growth in local agriculture, forestry, animal husbandry and fisheries.

## Rural transportation infrastructure and poverty

The relationship between rural transportation infrastructure and poverty remains complex. In terms of its direct impact, improvements in rural transportation infrastructure can reduce the transportation costs of farmers and expand the market opportunities for local agricultural products [20]. However, due to the remote location of and the lack of talent and capital concentration in rural areas, new roads transfer a large amount of labor to the city. This directly leads to the loss of local talent and the hollowing out of rural areas, which aggravates rural poverty [21, 22].

On the other hand, rural transportation infrastructure can reduce poverty indirectly by promoting tertiary industry and stimulating economic growth. The trickle-down effects from economic growth contribute to poverty alleviation. As one of the factors of production, investment in rural transportation infrastructure can promote the division of labor, improve production efficiency, and directly promote economic growth. The multiplier effect produced by investment can further stimulate the vitality of economic growth. In addition, as a public good, rural transportation infrastructure has externalities. This means that investment in rural transportation infrastructure can cause farmers to accumulate human capital and can promote manufacturing production. Additionally, transportation costs can consequently decrease, which can improve the overall investment structure to allow for additional investments and bring about economic growth [23–28]. Economic growth can reduce poverty by increasing employment opportunities and improving transfer payments, which are both trickle-down effects.

Therefore, we have hypotheses presented as below:

Hypothesis 2a. Rural transportation infrastructure may directly aggravate rural poverty.

Hypothesis 2b. Rural transportation infrastructure can promote tertiary industry growth and thus indirectly reduce poverty.

The mechanism can be pictured in Fig 1.

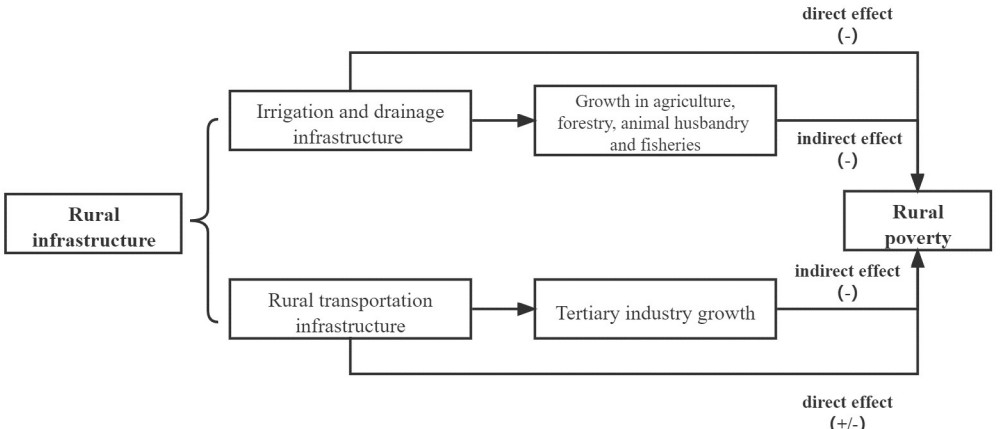

**Fig 1. The mechanism of irrigation and drainage and rural transportation infrastructure on poverty.** The direct an indirect mechanism of irrigation and drainage and rural transportation infrastructure on poverty.

## Materials and methods

### Data source

Due to limited data availability, this paper uses panel data on 26 provinces in China from 2002 to 2017. The data are collected mainly from the 2003–2018 China Rural Statistical Yearbook, China Statistical Yearbook, and the National Bureau of Statistics. The ethnic regions in this paper are eight ethnic provinces in China, referring to the Inner Mongolia Autonomous Region, Ningxia Hui Autonomous Region, Xinjiang Uygur Autonomous Region, Tibet Autonomous Region and Guangxi Zhuang Autonomous Region, as well as Guizhou, Yunnan and Qinghai provinces. Poverty is more serious in these areas for environmental and historical reasons. Additionally, due to a lack of data, we do not consider Beijing, Shanghai, Hainan, or Tibet.

### Methods

**Definition of variables.** *Dependent variables.* Rural poverty incidence rate (Poverty_rate) and the size of the rural poor population (Poverty_num). We define people in rural China whose income falls below the minimum income required by the government as the rural poor. The proportion of these people to the total rural population is the rural poverty incience rate. With the targeted poverty alleviation program in China, those who receive basic living allowances account for an increasing proportion of the impoverished population. They are good representatives for those who live in extreme poverty.

*Core independent variables.* Referring to the definition of [29], rural infrastructure is different from traditional public services. It caters to farmers' production, life and development with a long service life and two of the most important are irrigation and drainage and rural transportation infrastructure. Irrigation and drainage infrastructure is closely related to peasants' production and may help them increase family business income. While rural transportation infrastructure is closely related to non-agriculture employment and may help them increase wage income. Both irrigation and drainage and rural transportation infrastructure are crucial to poor peasants' income and poverty reduction. We choose effective irrigation area to represent the construction of irrigation and drainage infrastructure as it can reflect the actual

irrigation effect on cultivated land. And for rural transportation infrastructure, we choose rural road mileage to represent its construction.

*Intermediary variable*. According to the previous theoretical analysis, irrigation and drainage infrastructure can indirectly reduce poverty by promoting the growth of agriculture, forestry, animal husbandry and fisheries. Rural transportation infrastructure can indirectly reduce poverty by influencing the growth of the service industry [30]. Therefore, for irrigation and drainage infrastructure, we choose the added value of agriculture, forestry, animal husbandry and fisheries as the intermediary variable [31, 32]. While for rural transportation infrastructure, we choose and the added value of tertiary industries as intermediary variable [24].

*Control variables*. To improve the robustness of the model, this paper refers to previous studies [1, 12] and selects control variables from the three dimensions of economy, society and environment. The development of economy, society and environment may have influences on poverty through many ways. For example, economic development may reduce poverty through increasing consumption and expanding the channels of employment. And the development of society may affect the social welfare of the peasants and then pose impact on poverty. While the development of environment may directly influence the agricultural production and then affect welfare of the peasants and poverty. And referring to [9, 12, 27], the control variables at the economic level include per capita GDP, per capita industrial output, industrial structure, government expenditures, and rural residents' consumption levels. From the social dimension, the control variables are population density, human capital, population urbanization rate, land urbanization rate, urban-rural income gap, rural electricity consumption, mechanization level, and level of financial support for agriculture. From the environmental dimension, the control variables are land area available for crop planting, reservoir capacity, soil erosion control, and grain yield. The proxy indicators for each control variable are shown in Table 1.

**Difference-in-differences.** In 2010, to improve peasants' income and reduce poverty, the Chinese government issued No. 1 central document to strengthen investment to rural infrastructure especially irrigation and drainage infrastructure and rural transportation infrastructure. Therefore, this paper will use a difference-in-differences (DID) model and use the investment as a quasi-natural experiment. The impact of rural infrastructure on poverty will be evaluated in this way. This paper selects nonethnic areas as the experimental group and ethnic areas as the control group. The reasons are as follows. Due to natural resource endowments and historical developments, ethnic areas fall far behind nonethnic areas in economic and social terms. According to that situation in China and the theory of Development Poles, the investments in rural infrastructure in China is also developed-region-oriented. Referring to [32], DID applies to this case if the policy in 2010 has relatively larger effects on nonethnic areas than ethnic areas. Theoretically, the parallel trend assumption of DID is satisfied. We will test the assumption in the Results section. The model is as follows:

**Basic difference-in-difference model**.

$$\text{Poverty\_rate}_{it} = \alpha_0 + \alpha_1 \ \text{du}_{\text{nonethnic}} \cdot dt_{2010} + \sum \alpha_j \ \text{Control}_{it} + \lambda_i + \nu_t + \varepsilon_{it} \tag{1}$$

where Eq (1) represents the basic DID model. The dummy variable $du_{nonethnic}$ = 1 if belonging to *nonethnic* areas with better natural resource and faster development, and 0 otherwise. $dt_{2010}$ = 1 in the years after the policy stimulus in 2010. The coefficient, $\alpha_1$, therefore indicates the impact of rural infrastructure investment on rural poverty rate. The policy under study was implemented in 2010, which is the treatment period. $du_{nonethnic} \cdot dt_{2010}$ is the interaction between the group and time dummy variables; its coefficient denotes the net effect of policy

**Table 1. Definition of the variables in the model.**

| Variable type | Dimension | Variable | Symbol | Description |
|---|---|---|---|---|
| Dependent variable | Poverty | The size of the rural poor population | Poverty_num | The number of the rural poor population |
| | | Rural poverty incidence rate | Poverty_rate | Rural minimum living guarantee population / Total rural population |
| Independent variable | Rural infrastructure | Irrigation and drainage infrastructure | Irrigation | Effective irrigation area (1000 HA) |
| | | Rural transportation infrastructure | Road | Rural road mileage (km) |
| Intermediary variable | Industrial added value | The added value of agriculture, forestry, animal husbandry and fisheries | Primary industry | Added value of agriculture, forestry, animal husbandry and fishery (100 million yuan) |
| Control variable | Economy | The added value of tertiary industries | Tertiary industry | The added value of tertiary industries (100 million yuan) |
| | | Per capita GDP | Pgdp | Per capita GDP (yuan/person) |
| | | Per capita industrial output | Pindustrialization | Industrialization level (10000 yuan / person) |
| | | Industrial structure | Industrial structure | The ratio of the sum of the primary industry and the secondary and tertiary industries |
| | | Government expenditures | Government expenditures | Reflecting government expenditure (100 million yuan) |
| | | Rural residents' consumption levels | Consumption | Reflect the expenditure of rural residents (yuan / person) |
| | Society | Population density | Population | Number of people per unit land area (person / $km^2$) |
| | | Human capital | Human capital | Years of education per capita (years) |
| | | Population urbanization rate | Popu_urban | Proportion of urban population |
| | | Land urbanization rate | Land_urban | Built up area (10000 square kilometers) |
| | | Urban-rural income gap | Urban-rural_gap | Income of urban residents / Rural residents |
| | | Rural electricity consumption | Electricity | Rural electricity consumption (100 million kwh) |
| | | Mechanization level | Mechanization | Total power of agricultural machinery (10000 kW) |
| | | Level of financial support for agriculture | Afinance | Local expenditure on agriculture, forestry and water affairs (100 million yuan) |
| | Environment | Land area available for crop planting | Seed | Sown area of crops (1000 HA) |
| | | Reservoir capacity | Reservoir | Total reservoir capacity (100 million cubic meters) |
| | | Soil erosion control | Erosin | Soil erosion control area (1000 HA) |
| | | Grain yield | Grain | Grain yield per unit area (kg / HA) |

implementation, which is of great importance. $\lambda_i$ and $v_t$ represent the province fixed effects and the time fixed effects, respectively.

**Continuous difference-in-difference model**.

$$\text{Poverty\_rate}_{it} = \alpha_0 + \sum \alpha_m X_{it} \cdot dt_{2010} + \sum \alpha_j \text{ Control }_{it} + \lambda_i + v_t + \varepsilon_{it} \qquad (2)$$

In Eq (2), the group dummy variable $du_{nonethnic}$ is replaced by the continuous variable $X_{it}$, which represents the different types of infrastructure construction in this article. That is called continuous DID because $\alpha_m$ represents the net effect of the change in each rural infrastructure on poverty [33].

**Model of parallel trends**.

$$\begin{aligned} \text{Poverty\_rate}_{it} = \quad & \beta_0 + \beta_1 D_{it}^{-3} + \beta_2 D_{it}^{-2} + \beta_3 D_{it}^{-1} + \beta_4 D_{it}^{0} \\ & + \beta_5 D_{it}^{1} + \beta_6 D_{it}^{2} + \beta_7 D_{it}^{3} + \sum \alpha_j Z_{it} + \varepsilon_{it} \end{aligned} \qquad (3)$$

where the dummy variables of $D_{it}$ equal zero, except for the following: $D_{it}^{-j}$ equals 1 for experimental groups in the jth year before policy, while $D_{it}^{+j}$ equals 1 for experimental groups in the

jth year after policy. We examine the trends of poverty rates before and after 3 years of the policy. Also, we add the year of the policy and estimate the dynamic effect of policy on poverty rates relative to the year of policy. More importantly, we can test the parallel trend of DID. If $\beta_k$ (k = 1,2,3) before the policy is not significant, then the parallel trend assumption is satisfied. This means that the experimental group and the control group have similar trends before the policy. Similarly, if $\beta_k$ (k = 4,5,6,7) after the policy is significant, it means that differences arise between the experimental group and the control group after the policy implementation.

*Mediating effect model.* Referring to Wen et al.(2004) and Wen et al.(2014) and to test the influence mechanisms of different types of rural infrastructure on poverty, this paper further uses the mediating effect model [34, 35]. Eqs (4)–(6) are the regression equations set by the intermediary effect test procedure.

$$y_{it} = \beta + cX_{it} + \alpha \sum Z_{it} + \vartheta_t + \mu_t + \varepsilon_{it} \tag{4}$$

$$M_{it} = \beta + aX_{it} + \alpha \sum Z_{it} + \vartheta_t + \mu_t + \varepsilon_{it} \tag{5}$$

$$y_{it} = \beta + c'X_{it} + bM_{it} + \alpha \sum Z_{it} + \vartheta_t + \mu_t + \varepsilon_{it} \tag{6}$$

where the first step is to regress the dependent variable $y_{it}$ on the independent variable $X_{it}$ to confirm that $X_{it}$ is a significant predictor of $y_{it}$ in Eq (4). Then regress the mediator $M_{it}$ on $X_{it}$ to confirm that $X_{it}$ is a significant predictor of $M_{it}$ in Eq (5). Finally, regress $Y_{it}$ on both $X_{it}$ and $M_{it}$ to confirm that the $M_{it}$ is a significant predictor of $Y_{it}$ in Eq (6). The test procedure of mediating effect test is shown in Fig 2 [36]. The masking effect indicates $X_{it}$ may show no effect on $y_{it}$ on the whole, as positive and negative offsets each other. Fig 2 plots the test procedure of mediating effect.

## Results

### Descriptive statistical analysis

The descriptive statistics for the main variables are shown in Table 2. To further ensure the stability of the data and reduce problems such as collinearity and heteroscedasticity in the model, this paper conducts logarithmic transformation on the variables in the data except for the

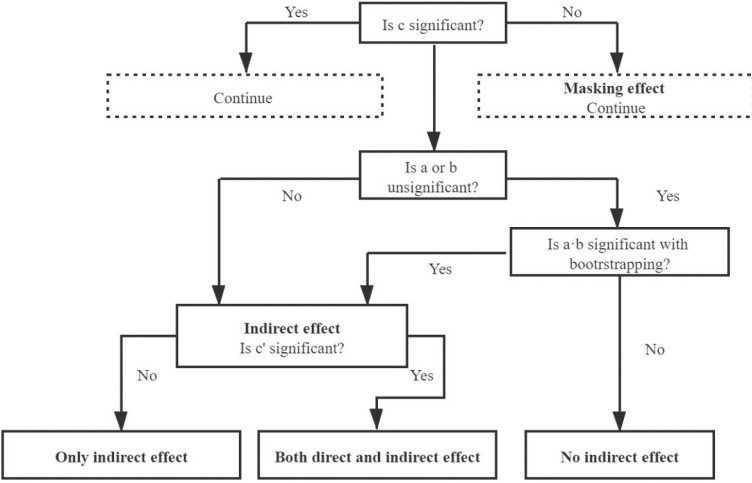

**Fig 2. Test procedure of mediating effect.** The 3 test procedure of mediating effect based on Eqs (4)–(6).

**Table 2. Descriptive of the variables in the model.**

| Variable | Obs | Mean | Std. Dev. |
|---|---|---|---|
| ln_Poverty_num | 399 | 1.409 | 0.426 |
| Poverty_rate | 407 | 0.062 | 0.051 |
| ln_Irrigation | 416 | 7.487 | 0.761 |
| ln_Road | 324 | 7.412 | 1.061 |
| ln_Primary industry | 416 | 6.986 | 0.944 |
| ln_Tertiary industry | 416 | 8.232 | 1.092 |
| ln_Pgdp | 416 | 10.06 | 0.72 |
| ln_Pindustrialization | 416 | -0.074 | 0.781 |
| ln_ Industrial structure | 416 | 0.143 | 0.06 |
| ln_ Gconsume | 416 | 7.192 | 0.921 |
| ln_ Rconsume | 416 | 8.438 | 0.673 |
| ln_ Population | 416 | 5.191 | 1.149 |
| Human_capital | 416 | 8.414 | 0.782 |
| Popu_urban | 409 | 0.474 | 0.105 |
| Land_urban | 416 | 7.018 | 0.78 |
| Urban-rural_gap | 416 | 2.995 | 0.58 |
| ln_ Electricity | 416 | 4.604 | 1.325 |
| ln_ Mechanization | 416 | 7.789 | 0.786 |
| ln_ Afinance | 416 | 5.146 | 1.136 |
| ln_ Seed | 416 | 8.501 | 0.727 |
| ln_ Reservoir | 416 | 5.273 | 0.839 |
| ln_ Erosin | 416 | 8.034 | 0.794 |
| ln_ Grain | 416 | 8.487 | 0.216 |

share of the population receiving the rural minimum living guarantee, industrial structure, human capital, population urbanization rate, and urban-rural gap. "ln" before a variable name indicates that logarithmic transformation has been carried out.

## Result of basic model

Using Eq (1), we first test the effect of rural infrastructure investment on poverty reduction and control for the fixed effects of provinces and years, as well as the control variables at the provincial level. First, following the benchmark from the DID model shown in Eq (1), the poverty incidence rate is taken as the explanatory variable. The effect is estimated by a multiway fixed effect model. Since the investment plan was promoted in 2010, this paper selects 2010 as the treatment period. The estimated results are shown in Table 3. In Table 3, Column (1) does not include control variables, while Column (2) does. The coefficient on the interaction between the time dummy variable and group dummy variable is the focus of our attention and reflects how rural infrastructure affects poverty. In Column (2), the interaction between the group dummy variable and the dummy variable for 2010 is significantly negative. This result suggests that rural infrastructure construction in 2010 effectively helped alleviate poverty in China.

However, this conclusion does not reveal irrigation and drainage infrastructure and rural transportation infrastructure can influence rural poverty. To address these questions, this article will further use the continuous DID model to investigate the effects of irrigation and drainage infrastructure and rural transportation infrastructure on poverty reduction. The test

**Table 3. Result of benchmark model of DID.**

| | (1) Poverty_rate | (2) Poverty_rate |
|---|---|---|
| duxd$t_{2010}$ | -0.0485*** | -0.0436*** |
| | (-4.71) | (-5.46) |
| Control for Year | Yes | Yes |
| Control for Province | Yes | Yes |
| Control Variables | No | Yes |
| Constant | 0.077*** | 1.62* |
| | (23.01) | (2.05) |
| Observations | 407 | 400 |
| R-squared | 0.8203 | 0.8791 |

Note:t statistics in parentheses,

* p <0.10,

** p <0.05,

*** p <0.01

results based on Eq (2) are shown in Table 4. As shown in this table, Columns (1) and (3) show the estimation result without control for variables. Added with control variables to increase the robustness of the results, Columns (2) and (4) reveal how irrigation and drainage infrastructure and rural transportation infrastructure influence poverty, respectively. Column (2) shows that the coefficient on interaction term between irrigation and drainage infrastructure and the year 2010 is negative and has a significance level of 5%. The results prove that irrigation and drainage infrastructure can effectively reduce poverty. In Column (4), the interaction coefficient between rural transportation infrastructure and the year 2010 is not statistically significant. This result indicates that unlike the other two types of rural infrastructure, rural transportation has no overall significant impact on poverty.

To check the robustness of above results, we display the result of parallel trend test, instrumental variable estimation and changing the dependent variable. Table 5 displays the parallel trend test results. Referring to [37], we choose the period three years before and after the implementation of the policy (2010) to test for common trends. As shown in Table 5, control variables are included in Columns (2) while not in Columns (1) to ensure the robustness of the regression results. Column (2) shows that the regression coefficients on the interaction terms

**Table 4. Result of continuous model of DID.**

| | (1) Poverty_rate | (2) Poverty_rate | (3) Poverty_rate | (4) Poverty_rate |
|---|---|---|---|---|
| ln_Irrigation×$t_{2010}$ | -0.00999 | -0.0177** | | |
| | (-1.66) | (-2.75) | | |
| ln_Rroad×$t_{2010}$ | | | 0.0075** | -0.0031 |
| | | | (2.21) | (-0.75) |
| Control for Year | Yes | Yes | Yes | Yes |
| Control for Province | Yes | Yes | Yes | Yes |
| Control Variables | | Yes | | Yes |
| Constant | 0.0954*** | 0.968 | 0.0456*** | 0.574 |
| | (4.69) | (0.96) | (3.05) | (0.54) |
| Observations | 407 | 400 | 311 | 311 |
| R-squared | 0.7818 | 0.8690 | 0.8367 | 0.8588 |

**Table 5. Parallel trend test.**

|  | (1) | (2) |
|---|---|---|
| Pre3 | -0.00679 | -0.0119 |
|  | (-0.97) | (-1.00) |
| Pre2 | -0.00649 | -0.00771 |
|  | (-0.83) | (-0.80) |
| Pre1 | -0.0108* | -0.00995 |
|  | (-1.75) | (-1.23) |
| Current | -0.0291 | -0.0240* |
|  | (-1.69) | (-1.72) |
| Aft1 | -0.0344** | -0.0242** |
|  | (-2.12) | (-2.28) |
| Aft2 | -0.0371** | -0.0269** |
|  | (-2.33) | (-2.35) |
| Aft3 | -0.0384** | -0.0239*** |
|  | (-2.67) | (-2.88) |
| Control for Year | YES | YES |
| Control for Province | YES | YES |
| Control Variables |  | YES |
| Constant | 0.0693*** | 0.441 |
|  | (20.01) | (0.45) |
| Observations | 407 | 400 |
| R-squared | 0.7656 | 0.8395 |

Note:t statistics in parentheses,

* $p < 0.10$,

** $p < 0.05$,

*** $p < 0.01$

between the time dummy variables and group dummy variables are not significant in 2007, 2008 or 2009. This finding shows that before the implementation of rural infrastructure construction in 2010, the incidence of poverty in ethnic areas and nonethnic areas experienced the same trend. The fluctuations in the two groups are not significantly different. The experimental group and the control group conform to the DID common trend assumption. Moreover, after 2010, the regression coefficients on the interaction terms between the time dummy variables and group dummy variables are significantly negative in 2011, 2012 and 2013. This result indicates that the trends in poverty in the experimental group and the control group were different after the implementation of the policy in 2010 and we cannot reject the test of parallel trends in pretreatment periods.

Fig 3 plots the impact of implementation of the policy on rural poverty rates. We consider a 6-year window, spanning from 3 years before the until 3 years after deregulation. The dashed lines represent 95% confidence intervals. Specifically, we report estimated coefficients in Eq (3). Fig 3 illustrates the same key points as Table 5: there is no trend in poverty rates in two groups prior to the policy. Next, note that poverty rates fall immediately after policy, such that Aft1 is negative and significant at the 5% level. Thus, the mechanisms and channels connecting rural infrastructure with the infrastructure must be fast acting.

Table 6 shows the result of instrumental variable estimation. The instrumental variables are lagged irrigation and drainage infrastructure and lagged rural transportation infrastructure. Columns (1) and (3) in Table 6 do not include the control variables and reflect the impact of

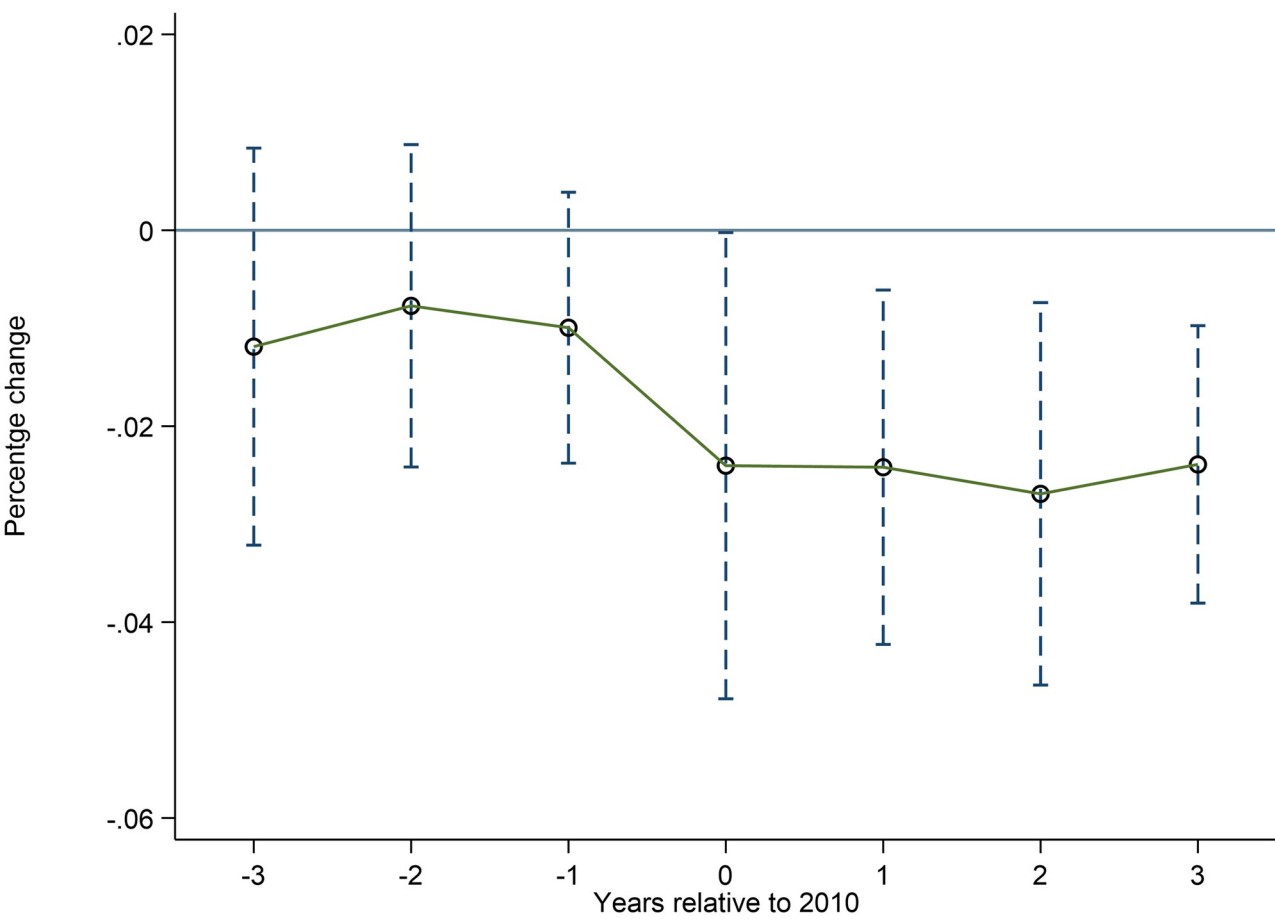

**Fig 3. The dynamic impact of rural infrastructure on poverty.** The impact of implementation of the policy on rural poverty rates 3 years before the until 3 years after deregulation.

**Table 6. Result of two-stage least square method.**

|  | (1) Poverty_rate | (2) Poverty_rate | (3) Poverty_rate | (4) Poverty_rate |
|---|---|---|---|---|
| ln_Irrigation×$t_{2010}$ | -0.00984*** | -0.0166*** |  |  |
|  | (-2.94) | (-4.89) |  |  |
| ln_Road×$t_{2010}$ |  |  | 0.00714*** | 0.0027 |
|  |  |  | (3.61) | (1.26) |
| Control for Year | YES | YES | YES | YES |
| Control for Province | YES | YES | YES | YES |
| Control Variables |  | YES |  | YES |
| Constant | 0.00385 | 0.771 | 0.0577*** | 0.262 |
|  | (0.77) | (1.46) | (19.01) | (0.46) |
| Observations | 382 | 378 | 286 | 286 |
| R-squared | 0.6715 | 0.8032 | 0.3844 | 0.5273 |

Note:t statistics in parentheses,

* $p < 0.10$,

** $p < 0.05$,

*** $p < 0.01$

**Table 7. Result of changing the dependent variable.**

| | (1) ln_Poverty_num | (2) ln_Poverty_num | (3) ln_Poverty_num | (4) ln_Poverty_num | (5) ln_Poverty_num | (6) ln_Poverty_num |
|---|---|---|---|---|---|---|
| du×$t_{2010}$ | -0.354*** | -0.168*** | | | | |
| | (-4.86) | (-2.92) | | | | |
| ln_Irrigation×$t_{2010}$ | | | -0.0309 | -0.0579* | | |
| | | | (-0.63) | (-1.89) | | |
| ln_Road×$t_{2010}$ | | | | | 0.0189 | 0.0119 |
| | | | | | (1.03) | (0.94) |
| Control for Year | | | YES | YES | YES | YES |
| Control for Province | | | YES | YES | YES | YES |
| Control Variables | | | | YES | | YES |
| constant | 1.527*** | -3.353 | 1.515*** | -6.417 | 1.483*** | -15.27** |
| | (62.85) | (-0.29) | (9.00) | (-0.60) | (18.41) | (-2.57) |
| Observations | 399 | 392 | 399 | 392 | 311 | 311 |
| R-squared | 0.6887 | 0.7559 | 0.6569 | 0.7532 | 0.7601 | 0.8589 |

Note:t statistics in parentheses,

* p <0.10,

** p <0.05,

*** p <0.01

irrigation and drainage infrastructure and rural transportation infrastructure on poverty, respectively. Columns (2) and (4) in Table 6 include the control variables to the models from Columns (1) and (3), respectively. In Column (2), the coefficient on the interaction terms between *ln_Irrigation* and dummy variable $t_{2010}$ is significantly negative, which is consistent with the above results. In Column (4), the coefficient on the interaction term between *ln_Road* and dummy variable $t_{2010}$ is not significant, which is also consistent with the previous results. Once again, irrigation and drainage infrastructure can promote poverty reduction. These results show that the findings from the above analysis are robust. However, the effect of rural transportation infrastructure on poverty reduction is not clear.

Table 7 presents the regression results of replacing rural poverty rate with the size of the rural poor population in Eqs (1) and (2). To enhance robustness, the logarithm of the rural poor population is taken. Column (1) shows the estimation results based on Eq (1). Column (2) includes control variables in the model from Column (1). Columns (3) and (5) are the estimation results from Eq (2), showing the impact of irrigation and drainage and rural transportation infrastructure on the size of the rural poor population, respectively. Columns (4) and (6) add control variables based on (3) and (5). As shown in Columns (2), (4) and (6) in Table 6, the dependent variable is replaced by the size of the rural poor population. The coefficients on interaction term between du and $t_{2010}$ is -0.168 and has a significance level of 5%. The coefficients on interaction term between ln_Irrigation and $t_{2010}$ is -0.0579 sand has a significance level of 10%. While the coefficients on interaction term between ln_Road and $t_{2010}$ is not significant. These results are consistent with the previous regression results.

## Result of mechanisms test

From the above regressions, we can see that irrigation and drainage infrastructure can promote poverty reduction, but their poverty reduction mechanisms need to be verified. Although rural transportation infrastructure has no significant impact on poverty, it remains to be seen

**Table 8. Result of mediating effect.**

| | Irrigation and drainage infrastructure | | | Rural transportation infrastructure | | |
|---|---|---|---|---|---|---|
| | (1) Poverty_rate | (2) ln_Primary industry | (3) Poverty_rate | (4) Poverty_rate | (5) ln_Tertiary industry | (6) Poverty_rate |
| ln_Irrigation×$t_{2010}$ | -0.0177*** | -0.0061 | -0.0171*** | | | |
| | (-5.45) | (-0.87) | (-5.37) | | | |
| ln_Primary industry | | | 0.0927*** | | | |
| | | | (3.84) | | | |
| ln_Road×$t_{2010}$ | | | | 0.0023 | 0.0142*** | 0.0056** |
| | | | | (0.96) | (3.77) | (2.40) |
| ln_Tertiary industry | | | | | | -0.2309*** |
| | | | | | | (-6.13) |
| Control for Year | YES | YES | YES | YES | YES | YES |
| Control for Province | YES | YES | YES | YES | YES | YES |
| Control Variables | YES | YES | YES | YES | YES | YES |
| constant | 1.128** | 1.89*** | 0.9534* | -0.495 | -9.265*** | -2.635*** |
| | (2.12) | (1.61) | (1.82) | (-0.69) | (-8.32) | (-3.48) |
| Observations | 400 | 400 | 400 | 311 | 311 | 311 |
| R-squared | 0.869 | 0.998 | 0.8724 | 0.884 | 0.9994 | 0.8989 |

Note:t statistics in parentheses,

* $p < 0.10$,

** $p < 0.05$,

*** $p < 0.01$

whether it has a "masking effect" on poverty. Using the mediating effect test model proposed by [35, 36], this section explores the internal influence mechanisms of the two types of rural infrastructure on poverty. The results are shown in Table 8.

In Table 8, Columns (1)-(3) examine the poverty reduction mechanisms behind irrigation and drainage infrastructure. The growth of agriculture, forestry, animal husbandry and fisheries is selected as the intermediate variable for irrigation and drainage infrastructure. The mediating effect test procedure is shown in Fig 1. As shown in Column (1) in Table 8, the construction of irrigation and drainage infrastructure has a significant negative impact on poverty, so we continue to carry out the intermediary effect test. In Column (2), the coefficient on irrigation and drainage infrastructure, which estimates the infrastructure's effect on the intermediary variable of rural economic growth, is not significant. In Column (3), the coefficient on irrigation and drainage infrastructure is significantly negative, while that on the growth of agriculture, forestry, animal husbandry and fisheries is significantly positive. Therefore, further bootstrap tests are needed, and the results are shown in Table 9. The coefficients on both the direct and indirect effects of irrigation and drainage infrastructure are negative. However, the confidence interval for the indirect effect after correction is [-.0016786, .001194]. The value 0 is included in the interval, indicating that the indirect effect is not significant. The confidence interval for the direct effect after correction is [-.0300072, -.0099262], which excludes 0.

In Table 8, Columns (4)-(6) are used to test the impact of rural transportation infrastructure on poverty. According to the regression results in Table 4, the overall impact of rural transportation infrastructure on poverty is not significant. However, how rural transportation infrastructure influences poverty has not been confirmed. Therefore, this section tests whether the transportation infrastructure has both indirect and direct impacts on poverty that create a masking effect. Due to the convenience brought by rural transportation infrastructure, on the

**Table 9. Result for bootstrapping.**

| | | Observed Coef. | Bias | Bootstrap Std. Err. | Normal-based [95% Conf. Interval] |
|---|---|---|---|---|---|
| ln_Irrigation×$t_{2010}$ | Indirect effect | -0.00057362 | -0.0001354 | 0.00082755 | [-.0021794, .0010257] (P) |
| | | | | | [-.0016786, .001194] (BC) |
| | Direct effect | -0.0171468 | 0.0004506 | 0.00400739 | [-.0248126, -.009654] (P) |
| | | | | | [-.0300072, .-.0099262] (BC) |
| ln_Road×$t_{2010}$ | Indirect effect | -0.0032976 | 0.0000615 | 0.00103574 | [.0053486, .0013547] (P) |
| | | | | | [.0056422, .0015639] (BC) |
| | Direct effect | -6.71096 | 0.0003991 | 0.00227134 | [.0015815, 0.0104943] (P) |
| | | | | | [.0003989, .0098849] (BC) |

(P) percentile confidence interval; (BC) bias-corrected confidence interval

one hand, the loss of the rural population leads to the hollowing out of rural areas and aggravates poverty. At the same time, the rural population mostly flows to the labor-intensive tertiary industry in cities and towns. This situation will lead to increases in wages and remittances to rural areas, reducing poverty. Therefore, this section selects the growth of the tertiary industry as the intermediary variable explaining how rural transportation infrastructure affects poverty. The results are shown in Table 8. The coefficient on rural transportation infrastructure in Column (4) is not significant. According to Fig 1, the impact of rural infrastructure on poverty may be masked. In Column (5), the coefficient on rural transportation infrastructure, which estimates its effect on the tertiary industry, is significantly positive. In Column (6), the coefficient of the growth of the tertiary industry on rural poverty is significantly negative, and the impact of rural transportation infrastructure on rural poverty is significantly positive. Table 9 displays the bootstrapping test results for rural transportation infrastructure. The results show that the corrected direct effect is significantly positive at the 95% level. The indirect effect is significantly negative at the 95% level.

## Discussion

This study develops a theoretical framework of how irrigation and drainage infrastructure and rural transportation infrastructure influence poverty. Along with provincial panel data on China from 2002 to 2017, this paper uses a basic DID model to examine how investment policy on rural infrastructure influences poverty in rural China. Then a continuous DID model is used to investigate how two of the most important rural infrastructures, irrigation and drainage infrastructure and rural transportation infrastructure, influence rural poverty in China. Instrumental variable estimation and a change in the independent variable are used to enhance the robustness of the results. In addition, the mediating effect model is used to investigate the influence mechanisms of irrigation and drainage infrastructure and rural transportation infrastructure on rural poverty in China.

This study found that rural infrastructure construction in 2010 can help reduce poverty in China. This study also revealed that irrigation and drainage infrastructure can directly reduce poverty, which is consistent with previous research [38–40]. Therefore, H1a is proved, and there is no evidence to support H1b. And there is no enough evidence for the indirect effect on poverty reduction in this study.

Besides, existing research on impact of rural infrastructure on poverty usually claims that rural infrastructure especially rural road may economic growth and therefore reduce poverty [41, 42]. This study found that rural transportation infrastructure may aggravates rural poverty

although it can reduce poverty by promoting the growth of the tertiary industry, which creates an "illusion" of no impact. Rural transportation infrastructure directly leads to rural hollowing out and aggravates rural poverty, indicating there is the siphon effect. In addition, rural transportation infrastructure promotes the growth of the tertiary industry, therefore indirectly promoting rural poverty reduction. Accordingly, the positive and negative effects offset each other, resulting in these effects being masked overall. Therefore, H2a and H2b are proved. These findings corroborate the ideas of Asher and Novosad(2020), who maintained that transportation infrastructure is costly investment and may not necessarily cause positive economic impacts [22].

Given above discussions, this article makes the following theoretical and empirical contributions to the literature as mentioned earlier: First, theoretically, we enrich literature on the poverty-reducing effect of irrigation and drainage infrastructure on poverty. Besides, this paper contributes to a wide literature by clarifying the poverty-aggravating and poverty-reducing and effect of rural transportation infrastructure and the underlying mechanisms. Second, empirically, the basic and continuous DID models and the mediating effect model are used to estimate the effects of rural infrastructure on poverty. Besides, two-stage least square method has been adopted to deal with the endogeneity problem, supplementing literature on estimating the causal relationship between rural infrastructure and poverty. Additionally, with updated and more extensive data, we empirically confirm the direct poverty-reducing effect of irrigation and drainage infrastructure and the direct poverty-aggravating and indirect poverty-reducing and effect of rural transportation infrastructure. These findings expand empirical research on the relationship between irrigation and drainage infrastructure and rural transportation infrastructure and poverty.

## Conclusions, implications and future research

### Conclusions

The final conclusions are as follows: First, irrigation and drainage infrastructure can effectively reduce rural poverty in China, and rural transportation infrastructure has no clear impact on poverty on the whole. Second, irrigation and drainage infrastructure can reduce poverty directly, and no evidence of indirect influence mechanisms has been found in this study. Third, for rural transportation infrastructure, it may bring about the siphon effect and directly aggravate rural poverty in China on the one hand. On the other hand, it can promote economic growth by driving the development of the tertiary industry and indirectly promotes poverty reduction in rural China through the trickle-down effects of economic growth. However, the two opposite effects offset each other, and rural transportation infrastructure has no obvious impact on poverty overall.

### Implications

Under the background of achieving the Sustainable Development Goals and realizing the aim of no poverty and facing the challenging of frequent extreme weather, this paper has clear policy significance for the global government on how to further strengthen the current poverty reduction effects of various rural infrastructure, consolidate the existing achievements in poverty alleviation and prevent poverty levels from increasing. In the context of urbanization, developing countries, represented by China, mostly focus on rural transportation to achieve poverty reduction. However, this paper finds that although rural transportation infrastructure can promote economic growth and achieve poverty reduction, it can also promote the decline of rural areas, and the poorest people are abandoned in rural areas. Moreover, the lack of irrigation and drainage infrastructure has a significant direct effect on poverty. In addition, under

the background that COVID-19 may exist for a long time around the world, some migrant workers have chosen to return to their hometown to engage in agricultural production and entrepreneurship, and even participate in the industrialization of agriculture. In this situation, the government is required to pay more attention to the construction of irrigation and drainage infrastructures which is of great importance to people's livelihood in rural areas. In addition, the government needs to increase financial input to poor rural communities to consolidate the achievements in poverty alleviation. Second, the reason why rural transportation infrastructure may aggravate poverty is that it can accelerate the outflow of the rural population. This means that although such infrastructure can promote urbanization, it may lead to rural hollowing out. However, urbanization is an inevitable trend in economic development worldwide. The government should, in combination with this trend, reposition rural functions, support local development projects with comparative advantages, and provide equal public services for the poorest rural populations.

## Limitations and future research

Although we have figured out the effect and mechanisms of irrigation and drainage and rural transportation infrastructure on poverty with empirical models, there are some limitations in this study. First, this article mainly focused on the irrigation and drainage and rural transportation infrastructure and did not consider other rural infrastructure such as electricity and telecommunications infrastructure, which are also important to poverty reduction. Future research can explore the impact and mechanism of electricity infrastructure on poverty. Second, the construction of rural infrastructure, especially transportation infrastructure, often has spatial spillover effects on nearby regions. The spillover effects were not considered in this article owing to limitations of paper length. Future study can consider both time and space effects of rural infrastructure on poverty by applying spatial Difference-in-Difference models [43, 44]. Third, this study only adopted panel data in different provinces during 2002–2017 due to data availability. Data covering a longer period and smaller units such as macro-county and even micro-household level can help provide a more accurate and convinced analysis on the impact of rural infrastructure on poverty, which can be studied in future research.

## Supporting information

**S1 Data.**
(DTA)

## Author Contributions

**Conceptualization:** Haitao Wu.

**Data curation:** Xiaodi Qin.

**Funding acquisition:** Haitao Wu.

**Investigation:** Tiecheng Shan.

**Methodology:** Haitao Wu.

**Project administration:** Haitao Wu.

**Supervision:** Haitao Wu, Tiecheng Shan.

**Writing – original draft:** Xiaodi Qin.

**Writing – review & editing:** Xiaodi Qin.

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
