## [Decision Letter · Decision Letter 0]

24 Jan 2022

PONE-D-22-00776Rural Infrastructure and Poverty in ChinaPLOS ONE

Dear Dr. Wu,

Thank you for submitting your manuscript to PLOS ONE. After careful consideration, we feel that it has merit but does not fully meet PLOS ONE’s publication criteria as it currently stands. Therefore, we invite you to submit a revised version of the manuscript that addresses the points raised during the review process.

We look forward to receiving your revised manuscript.

Kind regards,

Jun Yang

Academic Editor

PLOS ONE

Journal Requirements:

[NO authors have competing interests]. 

4. PLOS requires an ORCID iD for the corresponding author in Editorial Manager on papers submitted after December 6th, 2016. Please ensure that you have an ORCID iD and that it is validated in Editorial Manager. To do this, go to ‘Update my Information’ (in the upper left-hand corner of the main menu), and click on the Fetch/Validate link next to the ORCID field. This will take you to the ORCID site and allow you to create a new iD or authenticate a pre-existing iD in Editorial Manager. Please see the following video for instructions on linking an ORCID iD to your Editorial Manager account: https://www.youtube.com/watch?v=_xcclfuvtxQ.

Additional Editor Comments:

The authors estimated basic and continuous difference-in-differences (DID) models to investigate the preliminary impact

of irrigation and drainage infrastructure and rural transportation infrastructure on poverty and further explores the influence mechanisms of these rural infrastructures on poverty by using the mediating effect model. The topic is interesting. Authors are suggested to make the following changes to the paper.

The introduction should further highlight the scientific problems, motivations, and possible innovations of the paper.

It is suggested to update the data of the paper, so that the results may be more meaningful.

Discussion: The contribution of the paper should include both theoretical and empirical aspects, and the author is advised to rewrite

from these two aspects.

It should contain a summary of current research gaps and an outlook for future research.

The study examined the effects of rural transportation infrastructure on poverty using DID models and mediating effect model. The methods are sound, the analyses are reasonable, and the structure is complete. However, there are still some small problems that need to be fixed.

1. The citation format in the text needs to be checked and in accordance with the standard of the journal.

2. “This increase in poverty may offset the poverty reduction effect of rural infrastructure, which is called the masking effect of rural infrastructure.” (Line80-81) This sentence requires literature support.

3. The clarity of Figure 1-3 needs to be improved. There may be problems with Figure 1-2 displays that need to be checked.

Reviewers' comments:

Reviewer's Responses to Questions

**Comments to the Author**

1. Is the manuscript technically sound, and do the data support the conclusions?

Reviewer #1: Yes

Reviewer #2: Yes

2. Has the statistical analysis been performed appropriately and rigorously? 

Reviewer #1: Yes

Reviewer #2: Yes

3. Have the authors made all data underlying the findings in their manuscript fully available?

Reviewer #1: Yes

Reviewer #2: Yes

4. Is the manuscript presented in an intelligible fashion and written in standard English?

Reviewer #1: Yes

Reviewer #2: Yes

5. Review Comments to the Author

Reviewer #1: The study examined the effects of rural transportation infrastructure on poverty using DID models and mediating effect model. The methods are sound, the analyses are reasonable, and the structure is complete. However, there are still some small problems that need to be fixed.

1. The citation format in the text needs to be checked and in accordance with the standard of the journal.

2. “This increase in poverty may offset the poverty reduction effect of rural infrastructure, which is called the masking effect of rural infrastructure.” (Line80-81) This sentence requires literature support.

3. The clarity of Figure 1-3 needs to be improved. There may be problems with Figure 1-2 displays that need to be checked.

Reviewer #2: The authors estimated basic and continuous difference-in-differences (DID) models to investigate the preliminary impact

of irrigation and drainage infrastructure and rural transportation infrastructure on poverty and further explores the influence mechanisms of these rural infrastructures on poverty by using the mediating effect model. The topic is interesting. Authors are suggested to make the following changes to the paper.

The introduction should further highlight the scientific problems, motivations, and possible innovations of the paper.

It is suggested to update the data of the paper, so that the results may be more meaningful.

Discussion: The contribution of the paper should include both theoretical and empirical aspects, and the author is advised to rewrite

from these two aspects.

It should contain a summary of current research gaps and an outlook for future research.

6. PLOS authors have the option to publish the peer review history of their article (what does this mean?). If published, this will include your full peer review and any attached files.

Reviewer #1: No

Reviewer #2: No

---

## [Author Response · Author response to Decision Letter 0]

7 Mar 2022

Dear Editors and Reviewers,

We appreciate the opportunity to modify our manuscript according to the critical comments concerning our manuscript entitled “Rural Infrastructure and Poverty in China” (Manuscript Number: PONE-D-22-00776). We would like to thank the editors and the review team for your constructive comments and suggestions on our paper. They are all valuable and very helpful for revising and improving our paper, and simultaneously they are of great significance to give instructions to our research. We have studied the comments carefully and made revisions which we hope to meet with approval.

With a concerted effort to address all of the valuable advice, we have revised the paper to a more accurate and concise style including citation format and figures, highlight the scientific problems, motivations, and possible innovations, rewrite the theoretical and empirical contribution of the paper, provided additional information for the dataset, and have updated the referencing literature. We believe that the recommended changes have dramatically enhanced the manuscript’s quality and its contribution, and we hope you will agree.

Also, it should be noted that we highlight the important changes in yellow in our revised manuscript in pdf file. Below, we provide our detailed responses in tabular form to explain how your points have been included in the revision. As you will see, we have made every possible attempt to address your concerns, and we hope you will find our revision acceptable. Once again, thank you for helping us to improve the manuscript greatly.

Responses to Journal Requirements

Comment 1

Responses 1

Thanks for the guideline. The revised manuscript, as well as the title page, and supplementary materials are all arranged according to the PLOS ONE style.

Comment 2

Thank you for stating the following in your Competing Interests section: 

[NO authors have competing interests]. 

Responses 2

Thanks for the reminder. We have stated “The authors have declared that no competing interests exist” in the revised cover letter.

Comment 3

In your Data Availability statement, you have not specified where the minimal data set underlying the results described in your manuscript can be found. PLOS defines a study's minimal data set as the underlying data used to reach the conclusions drawn in the manuscript and any additional data required to replicate the reported study findings in their entirety. All PLOS journals require that the minimal data set be made fully available. For more information about our data policy, please see http://journals.plos.org/plosone/s/data-availability.

Upon re-submitting your revised manuscript, please upload your study’s minimal underlying data set as either Supporting Information files or to a stable, public repository and include the relevant URLs, DOIs, or accession numbers within your revised cover letter. For a list of acceptable repositories, please see http://journals.plos.org/plosone/s/data-availability#loc-recommended-repositories.

Any potentially identifying patient information must be fully anonymized.

Responses 3

Thanks for the reminder. Apology for our negligence. The minimal data set underlying the results described in your manuscript can be found at the website of National Bureau of Statistics of China (http://www.stats.gov.cn/), China Statistical Yearbook (https://data.cnki.net/yearbook/Single/N2021110004) and China Rural Statistical Yearbook (https://data.cnki.net/yearbook/Single/N2021120010). In addition, we have updated our Data Availability statement in the cover letter to reflect the information we provide. Besides, In the revised revision, we upload the study’s minimal underlying data set as Supporting Information files to improve the research repeatability.

Comment 4

PLOS requires an ORCID iD for the corresponding author in Editorial Manager on papers submitted after December 6th, 2016. Please ensure that you have an ORCID iD and that it is validated in Editorial Manager. To do this, go to ‘Update my Information’ (in the upper left-hand corner of the main menu), and click on the Fetch/Validate link next to the ORCID field. This will take you to the ORCID site and allow you to create a new iD or authenticate a pre-existing iD in Editorial Manager. Please see the following video for instructions on linking an ORCID iD to your Editorial Manager account: https://www.youtube.com/watch?v=_xcclfuvtxQ.

Responses 4

Thanks for the reminder. I have updated my information and fetched the ORCID iD. 

Responses to Reviewer #1

Comment 1

The citation format in the text needs to be checked and in accordance with the standard of the journal. 

Responses 1

Thanks a lot for the reminder. Apology for our negligence. In the manuscript we have revised the citation format adopted by PLOS. Besides, we also renewed the reference literature to prove the findings in this study and revised the reference style according to PLOS requirements, which has been highlighted in the revised manuscript. 

Please refer to Reference part in the revised manuscript for more details. 

Comment 2

“This increase in poverty may offset the poverty reduction effect of rural infrastructure, which is called the masking effect of rural infrastructure.” (Line80-81) This sentence requires literature support.

Responses 2

Thanks for this suggestion of supplementation. We feel sorry that we had neglected to provide enough literature support. We have modified this problem in the revised manuscript as follows:

“This increase in poverty may offset the poverty reduction effect of rural infrastructure, which we can call the masking effect, often used in mediation analysis in psychology [15–17].”

Please refer to Line 83-85, Page 3 in the revised manuscript for more details. 

Comment 3

The clarity of Figure 1-3 needs to be improved. There may be problems with Figure 1-2 displays that need to be checked.

Responses 3

Thanks for the suggestion for improving the clarity and displaying problems of Figure 1-3 of this manuscript. By applying the PACE tool, we have checked and revised Figure 1-3 in TIFF files according to figure preparation guidelines adopted by PLOS.

Please see in the revised Figures attached with manuscript.

References for Reviewer #1

1. MacKinnon, D. P., Krull, J. L., Lockwood, C. M. (2000). Equivalence of the mediation, confounding and suppression effect. Prevention science, 1(4), 173-181.

2. Lin, B. Q. (2003). Economic growth, income inequality, and poverty reduction in People's Republic of China. Asian development review, 20(2), 105-124.

3. Lu, H., Zhao, P., Hu, H., Zeng, L., Wu, K. S., Lv, D. (2022). Transport infrastructure and urban-rural income disparity: A municipal-level analysis in China. Journal of Transport Geography, 99, 103292.

Responses to Reviewer #2

Comment 1

The introduction should further highlight the scientific problems, motivations, and possible innovations of the paper.

Responses 1

Thank you very much for this constructive suggestion. During the revision stage, we tried to further highlight the scientific problems, motivations, and possible innovations of the paper in introduction. Motivated by China’s great investment to irrigation and transportation infrastructure especially in rural areas and the miracle of poverty reduction, we proposed the scientific problems cover whether the investment to irrigation and transportation infrastructure in 2010 reduced poverty in practice, what roles irrigation and transportation infrastructure played in influencing poverty and what is the underlying mechanisms. The possible innovations of the paper are highlighted from the theoretical and empirical aspects. We also add literature to support the viewpoint. The scientific problems, motivations, and possible innovations are enriched as following:

“Given that China has invested heavily to irrigation and transportation infrastructure especially in rural areas and the miracle of poverty reduction in the past 40 years, three important issues arise and needs to be figured out. First, did the investment to irrigation and transportation infrastructure in 2010 reduce poverty in practice? Second, what roles did irrigation and transportation infrastructure play in influencing poverty, respectively? Third, what is the mechanism behind the impact? Some studies maintain that rural transportation infrastructure, may not have social benefits and succeed in reducing poverty if rural laborers choose to transfer to urban areas due to the lack of agglomeration of local economies [2]. On the contrary, rural poverty may be deepened because of the loss of young and middle-aged rural labor force and the decline of abandoned villages, which is called siphon effect [3]. Moreover, irrigation and drainage infrastructure may also fail to count in that case. Dealing with these three problems can help China’s government reasonably adjust investment structure in rural infrastructure and consolidate the achievements on poverty reduction, as well as provide experience for those developing economies troubled by poverty. In addition, under the threat of extreme weather, it is of particular importance to clarify the role of irrigation and drainage infrastructure in affecting poverty. Irrigation and drainage infrastructure, due to the characteristics of relatively lower profits and slower return, usually attracts much less investment than rural transportation infrastructure. 

Given the above discussions, this research aims to evaluate the impact and mechanisms of rural irrigation and transportation infrastructure on poverty and makes innovations in several ways. First, this paper theoretically clarifies how irrigation and drainage infrastructure and rural transportation can affect poverty. We supplement theoretical literature concerning how irrigation and drainage infrastructure influences poverty and the underlying mechanism, which are rare in previous research. In addition, we add to a growing body of literature on how rural transportation influences poverty, focusing not on the poverty-reducing effect but also on the poverty-aggravating effect. Second, this paper helps enrich empirical evidence on how rural irrigation and transportation infrastructure influences poverty by applying the basic and continuous DID models and the mediating effect model. We use updated data to make the results more convincible and two-stage least square method to handle endogeneity problem.”

Please refer to Line 26-61, Page 2 in the revised manuscript for more details.

Comment 2

It is suggested to update the data of the paper, so that the results may be more meaningful.

Responses 2

Thanks a lot for your constructive suggestion of updating the data. We had tried to collect the latest data from the website of National Bureau of Statistics of China and China Statistical Yearbook, where authoritative data in China is provided. Unfortunately, one of the key control variables, Government expenditures, is only available by 2017 due to data availability. Therefore, data covering 2002-2017 is latest data we have access to. Besides, we think current panel data covering 16 years and 26 provinces is enough to support the findings of this study on how irrigation and drainage infrastructure and rural transportation affect poverty and the underlying mechanisms. But we are really grateful for your inspiring advice and we hope the data can updated in the future study so that the results may be more meaningful, which we highlight in the summary of current research gaps and an outlook for future research.

Please refer to Line 502-506, Page 16 in the revised manuscript for more details.

Comment 3

Discussion: The contribution of the paper should include both theoretical and empirical aspects, and the author is advised to rewrite from these two aspects.

Responses 3

Thanks for this suggestion. We have attempted to refine the contribution in discussion part, from the theoretical and empirical aspects, to present the importance of this research. The revised contribution is as following:

“Given above discussions, this article makes the following theoretical and empirical contributions to the literature as mentioned earlier: First, theoretically, we enrich literature on the poverty-reducing effect of irrigation and drainage infrastructure on poverty. Besides, this paper contributes to a wide literature by clarifying the poverty-aggravating and poverty-reducing and effect of rural transportation infrastructure and the underlying mechanisms. Second, empirically, the basic and continuous DID models and the mediating effect model are used to estimate the effects of rural infrastructure on poverty. Besides, two-stage least square method has been adopted to deal with the endogeneity problem, supplementing literature on estimating the causal relationship between rural infrastructure and poverty. Additionally, with updated and more extensive data, we empirically confirm the direct poverty-reducing effect of irrigation and drainage infrastructure and the direct poverty-aggravating and indirect poverty-reducing and effect of rural transportation infrastructure. These findings expand empirical research on the relationship between irrigation and drainage infrastructure and rural transportation infrastructure and poverty.”

Please refer to Line 438-452, Page 14 in the revised manuscript for more details.

Comment 4

It should contain a summary of current research gaps and an outlook for future research.

Responses 4

Thanks for this suggestion of supplementation. We feel sorry that we had neglected this part. We have modified this problem in the revised manuscript as follows: 

“Although we have figured out the effect and mechanisms of irrigation and drainage and rural transportation infrastructure on poverty with empirical models, there are some limitations in this study. First, this article mainly focused on the irrigation and drainage and rural transportation infrastructure and did not consider other rural infrastructure such as electricity and telecommunications infrastructure, which are also important to poverty reduction. Future research can explore the impact and mechanism of electricity infrastructure on poverty. Second, the construction of rural infrastructure, especially transportation infrastructure, often has spatial spillover effects on nearby regions. The spillover effects were not considered in this article owing to limitations of paper length. Future study can consider both time and space effects of rural infrastructure on poverty by applying spatial Difference-in-Difference models [43, 44]. Third, this study only adopted panel data in different provinces during 2002-2017 due to data availability. Data covering a longer period and smaller units such as macro-county and even micro-household level can help provide a more accurate and convinced analysis on the impact of rural infrastructure on poverty, which can be studied in future research.”

Please refer to Line 491-506, Page 15-16 in the revised manuscript for more details.

References for Reviewer #2

1. Xu, Z., Sun, T. (2021). The Siphon effects of transportation infrastructure on internal migration: evidence from China’s HSR network. Applied Economics Letters, 28(13), 1066-1070.

2. Donaldson, D. (2018). Railroads of the Raj: Estimating the impact of transportation infrastructure. American Economic Review, 108(4-5), 899-934.

3. Lipton, M., Litchfield, J., Faures, J. M. (2003). The effects of irrigation on poverty: a framework for analysis. Water policy, 5(5-6), 413-427.

4. Unver, O., Wahaj, R., Lorenzon, E., Mohammadi, K., Osias, J. R., Reinders, F., ... Sangjun, I. M. (2018). Key and smart actions to alleviate hunger and poverty through irrigation and drainage. Irrigation and Drainage, 67(1), 60-71.

5. Choi, K. S., Labhsetwar, V. K. (2021). Sustainable agricultural growth for rural development in Asia: a review. Irrigation and Drainage, 70(3), 470-478.

6. Aschauer, D. A. (1989). Is public expenditure productive? Journal of monetary economics, 23(2), 177-200.

7. Ke, X., Chen, H., Hong, Y., Hsiao, C. (2017). Do China’s high-speed-rail projects promote local economy? —New evidence from a panel data approach. China Economic Review, 44, 203-226.

8. Dube, J., Legros, D., Th´eriault, M., Des Rosiers, F. (2014). A spatial difference-in-differences estimator to evaluate the effect of change in public mass transit systems on house prices. Transportation Research Part B: Methodological, 64, 24-40.

9. Delgado, M. S., Florax, R. J. (2015). Difference-in-differences techniques for spatial data: Local autocorrelation and spatial interaction. Economics Letters, 137, 123-126.

---

## [Decision Letter · Decision Letter 1]

14 Mar 2022

PONE-D-22-00776R1Rural Infrastructure and Poverty in ChinaPLOS ONE

Dear Dr. Wu,

Thank you for submitting your manuscript to PLOS ONE. After careful consideration, we feel that it has merit but does not fully meet PLOS ONE’s publication criteria as it currently stands. Therefore, we invite you to submit a revised version of the manuscript that addresses the points raised during the review process.

We look forward to receiving your revised manuscript.

Kind regards,

Jun Yang

Academic Editor

PLOS ONE

Journal Requirements:

Additional Editor Comments (if provided):

Minor Revision

Reviewers' comments:

Reviewer's Responses to Questions

**Comments to the Author**

1. If the authors have adequately addressed your comments raised in a previous round of review and you feel that this manuscript is now acceptable for publication, you may indicate that here to bypass the “Comments to the Author” section, enter your conflict of interest statement in the “Confidential to Editor” section, and submit your "Accept" recommendation.

Reviewer #1: All comments have been addressed

Reviewer #2: (No Response)

2. Is the manuscript technically sound, and do the data support the conclusions?

Reviewer #1: Yes

Reviewer #2: (No Response)

3. Has the statistical analysis been performed appropriately and rigorously? 

Reviewer #1: Yes

Reviewer #2: (No Response)

4. Have the authors made all data underlying the findings in their manuscript fully available?

Reviewer #1: Yes

Reviewer #2: (No Response)

5. Is the manuscript presented in an intelligible fashion and written in standard English?

Reviewer #1: Yes

Reviewer #2: (No Response)

6. Review Comments to the Author

Reviewer #1: All problems have been addressed. It is recommended for publication.

Reviewer #2: The authors have revised their manuscript based on the review comments.

However, the format of the main text should be coherent with the requirement of Plos One.

7. PLOS authors have the option to publish the peer review history of their article (what does this mean?). If published, this will include your full peer review and any attached files.

Reviewer #1: No

Reviewer #2: No

---

## [Author Response · Author response to Decision Letter 1]

18 Mar 2022

Thank you very much for your appreciating the manuscript. As you suggested, we have used the LaTeX template provided by PLOS and compiled the LaTeX files as PDF to ensure the format of the main text should be coherent with the requirement of Plos One.

Besides, we have revised the reference style as outlined in the ICMJE sample references, which is required by Plos One.

---

## [Decision Letter · Decision Letter 2]

23 Mar 2022

Rural Infrastructure and Poverty in China

PONE-D-22-00776R2

Dear Dr. Wu,

We’re pleased to inform you that your manuscript has been judged scientifically suitable for publication and will be formally accepted for publication once it meets all outstanding technical requirements.

Kind regards,

Jun Yang

Academic Editor

PLOS ONE

Additional Editor Comments (optional):

Accept

Reviewers' comments:

Reviewer's Responses to Questions

**Comments to the Author**

1. If the authors have adequately addressed your comments raised in a previous round of review and you feel that this manuscript is now acceptable for publication, you may indicate that here to bypass the “Comments to the Author” section, enter your conflict of interest statement in the “Confidential to Editor” section, and submit your "Accept" recommendation.

Reviewer #1: All comments have been addressed

Reviewer #2: (No Response)

2. Is the manuscript technically sound, and do the data support the conclusions?

Reviewer #1: Yes

Reviewer #2: (No Response)

3. Has the statistical analysis been performed appropriately and rigorously? 

Reviewer #1: Yes

Reviewer #2: (No Response)

4. Have the authors made all data underlying the findings in their manuscript fully available?

Reviewer #1: Yes

Reviewer #2: (No Response)

5. Is the manuscript presented in an intelligible fashion and written in standard English?

Reviewer #1: Yes

Reviewer #2: (No Response)

6. Review Comments to the Author

Reviewer #1: It is recommended for publication.

Reviewer #2: The authors have adequately addressed comments raised in a previous round of review and I feel that this manuscript is now acceptable for publication.

7. PLOS authors have the option to publish the peer review history of their article (what does this mean?). If published, this will include your full peer review and any attached files.

Reviewer #1: No

Reviewer #2: No

---

## [Editor Report · Acceptance letter]

4 Apr 2022

PONE-D-22-00776R2 

Rural Infrastructure and Poverty in China 

Dear Dr. Wu:

I'm pleased to inform you that your manuscript has been deemed suitable for publication in PLOS ONE. Congratulations! Your manuscript is now with our production department. 

Kind regards, 

on behalf of

Dr. Jun Yang 

Academic Editor

PLOS ONE